# Influence of Lexical Development on Reading and Spelling Skills: Effects of Enhancement on Second-Grade Children in Primary School

**DOI:** 10.3390/children10081416

**Published:** 2023-08-19

**Authors:** Oriana Incognito, Alice Mercugliano, Lucia Bigozzi

**Affiliations:** Department of Education, Languages, Interculture, Literature and Psychology, University of Florence, 50121 Florence, Italy; alice.mercugliano@unifi.it (A.M.); lucia.bigozzi@unifi.it (L.B.)

**Keywords:** lexical competence, spelling accuracy, lexical enhancement program, reading skills, reading comprehension, primary school children

## Abstract

Previous studies suggest that lexical competence is an important factor that influences reading skills and spelling accuracy in primary school children. Understanding the relationship between these skills will provide valuable insights to improve reading and writing enhancement and intervention strategies. The aim of this pre-post longitudinal study is to examine the effectiveness of an enhancement program, in which there are activities proposed through a narrative and metacognitive methodology, designed to develop the cognitive processes of lexical acquisition and its effects on reading and writing ability. A total of 74 primary school children (M-age = 7.04 years) participated in the research. They were divided into groups: experimental, which carried out the enhancement, and control groups, which carried out the typical school program. The results show that children who carried out the enhancement obtained higher scores in reading skills, specifically in reading accuracy and text comprehension and spelling accuracy, in comparison with their peers in the control group. These results suggest that strengthening the lexical semantic pathway, as theorized by Coltheart’s two-way model, can lead to improved reading comprehension and diminished reading errors and spelling inaccuracies.

## 1. Introduction

Reading and spelling are fundamental skills that primary school children develop during their early years of education, and lexical competence is fundamental for language learning, but it is also an essential element for acquiring reading, writing, and communication abilities. Understanding how these skills are related can provide researchers and teachers with insights into reading and writing learning as well as better ways to improve lexical competence. Previous studies [1,2] have shown significant positive effects of an enhancement program, focused on the reinforcement of lexical competence, in Italian-speaking children with a mean age of nine years old. The results highlighted that increasing the number of words stored in the lexical storehouse, through the acquisition of complex lexical modes, leads to improved verbal fluency, comprehension, accuracy, and speed in reading, as well as spelling accuracy. These studies highlight the close relationship between lexical competence and encoding and decoding skills. In the study by Bigozzi and Biggeri [2] by means of a multivariate analysis using graphical models (Figure 1), it can be seen that the enhancement of lexical competence has a direct effect on the decrease of errors made in reading and writing tasks. This was due to an increase in efficiency in the semantic-lexical pathway according to Coltheart’s theorization.

The complex connections between lexical competence and reading skills or reading comprehension are confirmed by recent literature. Subjects with a strong vocabulary foundation find it easier to comprehend written texts, infer meanings from context, and express their ideas in writing and also have a tendency to show better reading comprehension skills [3,4]. In particular, several studies examine the relationship between lexical quality and reading skills in primary school children, which is important for its implication on academic learning. In the cross-sectional study by Richter et al. [5] on German primary school children (from 1st to 4th grade), results have shown that lexical quality, operationalized as the efficiency and accuracy of orthographic, phonological and meaning representations, helped to explain interindividual variation in text comprehension ability. In another study on the lexical quality hypothesis, results from Protopapas et al. [6] have shown considerable and stable correlations over time between vocabulary ability and comprehension in Greek children attending 2nd, 3rd, and 4th grades. Furthermore, this study has shown, through hierarchical regression analyses, how lexical quality mediates the effect of decoding on comprehension. Finally, one study [7] on 10-year-old children demonstrated that phonological skills, but also broader language skills, such as vocabulary, morphology, sentence correction, and sentence processing, predicted reading abilities, including speed word reading, non-word reading, spelling, and reading comprehension.

On the other hand, regarding the relationship between lexical competence and spelling abilities, different studies have shown their connection in samples who are learning English as a second language [8,9] or in adult samples [10]. In these studies, the importance of breadth, depth, and productivity of lexical competence in their interaction with writing skills has been confirmed, highlighting how this competence is a good predictor of the quality of writing performance. Actually, the novelty of studies already mentioned by Bigozzi and colleagues [1,2] lies in their emphasis on how lexical competence is fundamental to the development of spelling ability in monolingual and typically developing children. They investigated these aspects in a sample of primary school children with an average age of about nine years, in a language with transparent orthography, such as Italian.

Moreover, lexical competence has often been regarded as a natural process that begins with the pronunciation of first words and continues throughout the developmental stages; according to the incidental learning conception, this competence and vocabulary enrichment before entering school is strictly dependent on chance opportunities provided by listening and speaking situations within the family context, and after the formal literacy period, is strictly connected to the teaching modalities and practice [11]. However, given the fundamental role of this competence in reading and comprehension skills [4,12], educational research emphasizes that it is not appropriate to rely on incidental opportunities to develop this competence. For some time now, studies have highlighted the need to use intentional intervention even in early developmental periods, proposing a didactic setup following the cognitive processes of lexical acquisition [13,14,15,16].

For this reason, an enhancement program was designed to promote lexical competence in primary school children in a language with transparent orthography (such as the Italian language). Its implementation is based on Boschi, Aprile, and Scibetta’s [17] Multidimensional Model of Lexical Representation (MMLR), which outlines the different developmental processes involved in lexical expansion. According to this model, lexical expansion follows certain regularities as children grow: initially, they rely on less sophisticated defining modalities, such as tautologies, graphophonemic constraints, image value, frequency of use, and consecutive effects, which are associated with concrete and superficial aspects of words; as they mature, more complex and adult-like skills emerge, including categorizations, functionalizations, synonyms, antonyms, and the ability to understand the meaning of homonyms and polysemes within the context.

This pre-post longitudinal study aims to investigate whether the enhancement of the semantic-lexical pathway, as theorized by Coltheart, can also be extended to a sample of typically developing children attending the second primary grade. It was deemed appropriate to choose this age group for the development of lexical competence precisely to investigate what happens in the very early stages of learning written language. When children attend the first two years of primary school, they still exhibit those spelling and reading errors of the less developed type because they have not yet automated the competence of reading and writing. The theoretical criteria for justifying this age of interest lies in Coltheart’s dual route model [18,19] and in Firth’s staged model of the development of reading skills [20]. This staged model [20] suggests that at around four or five years old, children with a transparent language predominantly rely on the grapheme-phoneme conversion mechanism for word recognition, in line with Coltheart’s sub-lexical route. From the age of eight, the visual orthographic and phonological lexicon are consolidated, leading to more proficient reading and writing as the child’s lexical-semantic storehouse expands with new words. The overall idea is to examine whether Coltheart’s theorized semantic-lexical pathway enhancement is applicable to typically developing children of a specific age group (i.e., second-grade primary school children) and if it might have an impact on their reading strategies compared to the more effective sub-lexical route they tend to use according to previous studies [20].

### Rationale and Aims

Despite the importance lexical competence has, in Italian school systems the teaching methods generally used to enhance lexical competence and promote vocabulary enrichment are structured with memorization activities of specific words, focusing more on quantitative enrichment, that pertains to an increase in the quantity or number of words known, but not of qualitative enrichment. Qualitative enrichment is a key aspect in promoting vocabulary and lexical development, as it relates to a deeper and more comprehensive understanding of words, which allows one to reflect on their meanings, nuances, and use in different contexts and to promote the acquisition of fundamental processes of vocabulary expansion, such as categorization, functionalization, synonymy, and antonymy.

In the enhancement program used in this longitudinal study, using a narrative and metacognitive methodology, various activities are proposed that allow for the development of procedural lexical competence and deep reflection on words.

In light of these theoretical foundations, the main aim of this pre-post longitudinal study was to test the effectiveness of a lexical enhancement program on second-grade primary school children, in a language with transparent orthography (such as the Italian language). Specifically, we wanted to test whether the enhancement program was effective for:

(1) Reading skills, more specifically on text comprehension, reading time, and reading accuracy, in terms of fewer errors made in reading the assigned text;

(2) Writing skills, specifically on spelling accuracy, in terms of fewer errors made in writing dictation of sentences.

## 2. Materials and Methods

### 2.1. Participants

A total of 74 children (mean age (SD) = 7.04 (0.39); 42 male and 32 female) attending 2nd grade of a primary school in two mid-size towns in central Italy participated in the research. The participants were divided into two groups: experimental (*n* = 34) and control (*n* = 40). The participants were selected by balancing the classes: for each school, an experimental class and a control class. The experimental group carried out an enhancement program on lexical and spelling skills (see Section 2.2) and, at the same time, the control group followed routine teaching according to a schedule compiled by teachers and in accordance with ministerial programs, which also included some semantic exercises.

In Italy, primary school is compulsory, lasts five years, and is part of the first cycle of education. Children attending the second grade of primary school have an average age of 7 years.

According to national guidelines, until the end of the second grade of primary school, teachers focus primarily on the spelling component of writing, because only at the end of school are second graders expected to have finalized the acquisition of orthography [21].

Background information about home characteristics and sociocultural-economic status was collected using a parental questionnaire attached to the informed consent sheet. Specifically, information relating to the subjects’ environment was collected, for example, the place of birth and the level of education of both parents, to verify the homogeneity of the groups in terms of socio-environmental characteristics. The collection of information showed that the subjects have socio-environmental characteristics of the middle class, and coming from the same catchment area, belong to a homogeneous population.

Principals and teachers previously agreed with the aims and procedures of this study. The measures were administered at a time agreed upon with the school and with due adherence to the requirements of privacy and informed consent required by Italian law (Legislative Decree DL-196/2003). Regarding the ethical standards for research, the study referred to the latest version of the Declaration of Helsinki [22].

### 2.2. Measures and Enhancement Program

All children were asked to individually perform a battery of reading tests, namely the MT reading tests [23], and a spelling task extracted from the Battery for the Assessment of Writing and Spelling Proficiency [24]. All tests were administered in schools, during school hours, by a trained experimenter after scheduling meetings with the school principal and teachers involved.

**MT reading tests.** To assess text comprehension and reading skills, in terms of rapidity and correctness of decoding, the standardized battery of MT reading tests by Cornoldi and Colpo [23] was administered. The battery was specifically designed for Italian students. The MT Test has several standardized versions, one for each grade of Italian compulsory school and one for high school. For psychometric parameters, please refer to the manual [23].

Specifically, to test text comprehension, children were asked to silently read an expository text and answer 10 multiple-choice questions, choosing one of four possible answers. Through questions, students are asked, for example, to capture the literal meaning of a sentence, then be able to paraphrase a concept or draw semantic or lexical inferences. There were no time limits; moreover, in order to minimize the memory load, the children were allowed to return to the text whenever they wanted while answering the multiple-choice test. The administration lasted 60 min, which corresponds to one hour of school lessons. The final score was calculated as the total number of correct answers for the text read and ranged between 0 and 10.

Regarding reading skills, each child was asked to read the text aloud as best as he/she could, while the researcher noted the reading time and errors; as suggested by the manual [23], the reading time for each subject was 240 s maximum. This test produces two scores for reading accuracy and speed. Reading accuracy takes into account the number of errors made by the children while reading aloud, that is, mispronunciations, omitted words, or added syllables, as well as pauses longer than 5 s. Each type of error was counted only once during the test.

Reading speed refers to the ratio of time that the student takes to read the text (in seconds) and the total number of syllables read. Hence, the slower the children read, the higher the score.

**Battery for the Assessment of Writing and Spelling Proficiency.** To assess children’s spelling skills, in terms of the correctness of coding, the subtest of sentence dictation of the Battery for the Assessment of Writing and Spelling Proficiency [24] was used. Children were asked to write down 8 sentences dictated by the experimenter. The administration lasted 60 min, which corresponds to one hour of school lessons. The correctness score was calculated based on the number of errors made during dictation, balanced for the total number of written words, i.e., the ratio of the number of errors made to the number of words written. One point is counted for each error, if the same word contained more than 1 it was counted for the number of errors actually made. Some examples of errors were: grapheme swapping, omission or addition of letters or syllables, inversions, incorrect grapheme, illegal separations, illegal mergers, grapheme swap, omission or addition of «h», omission or addition of accents, omission or addition of double letter. Two independent raters coded the errors, the agreement between the raters was 98%, and the disagreements were discussed and resolved. The percentage of the inter-rater agreement was calculated according to the following formula: (number of concordant responses/total number of responses) * 100 [25].

**Enhancement program.** Children in the experimental group carried out a specific enhancement program on lexical competence that promotes the development and stimulation of vocabulary-building processes according to the Multidimensional Model of Lexical Representation [17].

The enhancement program was carried out during the school year for 6 months on a regular basis of once a week for about 2–3 h by the teacher for the experimental group.

The enhancement is implemented through an educational path of increasing difficulty, in which some well-known fairy tales are presented (e.g., Little Red Riding Hood). The tales have a simple rhyming structure and the “hot words”, on which the exercises are carried out, are highlighted (with different print formats). Children follow the reading of the story in a book, with the help of the pictures, and then perform the exercises proposed by cards.

Enhancement activities are structured in the form of worksheets:

(1) Tabs for overcoming assimilative tendencies: the purpose of this section is to help the children in overcoming the defining modes characteristic of childhood (“assimilative tendencies”). In the early years of primary school, we see a gradual, but rapid, abandonment of these word-generating mechanisms and the emergence of a mode of lexemic organization increasingly like that of adults. The criterion on which this stage of teaching action is based is to make constructive use of those defining processes that are commonly considered errors and are instead progressive steps in the process of growing lexical competence.

(2) Tabs for vocabulary enrichment: the purpose of this section is to promote in the student the acquisition of four fundamental processes of vocabulary expansion: categorization, functionalization, synonymy, and antonymy. Defining words by including them in categories is to be considered one of the earliest and most relevant processes in the organization of knowledge and experience. Enhancement helps children to define words by organizing them hierarchically into categories, with higher-grade groupings (superordinate categories: “Dogs are animals”) and lower-grade groupings (subordinate categories: “A Dalmatian is a dog”). Children are also stimulated to define words according to the dynamic-functional characteristics of objects; they are therefore invited to identify, on the basis of their own experiences, the functions of objects and the actions that can be performed on them or with them.

(3) Tabs for the enrichment of contextualization skills: the purpose of this section is to qualitatively improve children’s lexical heritage through knowledge of the rules that connect words in reciprocal relationships. Therefore, children are stimulated to use context, not only to understand the meaning of new words but also and especially to understand the different meanings of the same word.

Overall, through the use of tabs (Appendix A) and associated song labels (for words that have certain spelling obstacles, such as doubles, or specific spelling rules, such as the use of h, or contextualized expressions or idioms), children can work on the word’s meaning and form by writing, drawing, choosing, and discussing with a classmate the word’s meaning and play on the phonological-semantic intertwining. At the end of each tab, a suggestion is proposed to promote reflection on the activity carried out; the metacognitive calendar is also compiled and the newly learned words are written. At the end of each work section, a metacognitive path of evaluation and self-assessment is completed, which allows children to reflect on their own learning methods and on the reason that made it easy or difficult to tackle the exercises.

### 2.3. Research Design

This pre-post longitudinal study lasted one school year, from the beginning to the end of second grade of primary school.

The procedure involved administering the tests at two times, for the experimental group, before and after the implementation of the enhancement program and for the control group, after the same period without the administration of the enhancement (i.e., the participants were performing normal school activities and traditional exercises in Italian language). The initial assessment was scheduled in November, in which the entire specific assessment protocol was administered, including various tests and batteries designed to investigate various skills related to reading and spelling. At the end of the strengthening program (May-June, end of the school year), the same tests used initially were administered to each child for a post-test assessment.

## 3. Results

Table 1 shows the main descriptive statistics for the variables examined (text comprehension, reading inaccuracy, reading time, and spelling inaccuracy) in both groups (experimental and control). Levene’s test was used to assess the equality of variances. Specifically, regarding reading time, a higher mean indicates slower reading performance (see coding in the Section 2); regarding reading and spelling inaccuracy, it refers to the number of errors that children made in the two tasks.

Moreover, Table 2 shows the ANOVA statistics for comparison between experimental and control groups at the first administration (Time 1). The results show that there are no significant differences between experimental and control groups at Time 1, therefore participants have the same level at baseline.

A series of repeated ANOVA measures with a Greenhouse–Geisser correction was used to determine whether the enhancement program had an effect over time on text comprehension, reading inaccuracy, reading time, and spelling inaccuracy, within and between groups. Partial eta-square (η^2^) was computed to provide effect size in case of significance. Following Cohen [26] and Miles and Shevlin [27], the thresholds of partial η^2^ adopted in this study are small partial η^2^ > 0.01, medium > 0.06, large > 0.14.

First, for what concerns text comprehension (Figure 2), the results of within-subject contrasts show that there is a significant interaction effect between time and groups (time * groups: F(1,67) = 9.47, *p* < 0.01, η^2^ = 0.12). Specifically, the experimental group significantly improved performance at time 2, after performing the enhancement program. Whereas, no main effects were found for either time (within subjects) or group (between subjects).

Second, for reading inaccuracy (Figure 3), the results of within-subject contrasts show that there is both a main effect of the time (F(1,67) = 28.89, *p* < 0.001, η^2^ = 0.31)) and a significant interaction effect between the time and the groups (time*groups: F(1,67) = 36.34, *p* < 0.001, η^2^ = 0.35). Moreover, the results of contrasts between subjects show that there is a main effect of the groups on reading inaccuracy (F(1,67) = 5.14, *p* < 0.05, η^2^ = 0.07). These results suggest that after the enhancement program, the performances of the experimental group improved. There was indeed a significant decrease in the number of errors made in the reading task compared to their peers in the control group.

Third, for reading time (Figure 4), the results of within-subject contrasts show only a main effect of time (F(1,67) = 119.48, *p* < 0.001, η^2^ = 0.64). There are no significant interaction effects nor main effects of the group (between subjects). This means that independently of the experimental condition, the two groups improve their performance in reading time.

Finally, for spelling inaccuracy (Figure 5), the results of within-subject contrasts show that there is both a main effect of the time (F(1,66) = 72.71, *p* < 0.001, η^2^ = 0.52) and a significant interaction effect between the time and the groups (time * groups: F(1,66) = 8.68, *p* < 0.01, η^2^ = 0.12). However, there is no significant main effect of the group (between subjects). Therefore, these results show that the experimental group that carried out the enhancement program benefited the most, significantly decreasing the number of errors made in the spelling task.

## 4. Discussion

The main aim of this pre-post longitudinal study was to verify if a specific lexical enhancement program improved reading and spelling skills but also the reading comprehension of second-grade primary school children, as already demonstrated in previous studies conducted on older children [1,2].

The program led to improved reading comprehension and to diminished reading errors and spelling inaccuracies, which can be explained by the strengthening of the semantic-lexical pathway, as theorized by Coltheart’s two-way model [18,19].

It is possible to theorize that the improvement in reading comprehension and the decrease in reading errors in the experimental group is based on the fact that for the children following the enhancement program the semantic-lexical pathway is more easily accessed, with an increase in the number of words known of which the correct orthographic form and the semantic value are also stored. By going to act directly on the semantic-lexical pathway, the enhancement makes the recognition of words with direct access to the vocabulary faster and smoother and allows the reader to recover an articulatory pattern, permanently stored in the lexicon thanks to the “contextualized labels” of the words within the stories proposed in the treatment. Furthermore, the improvement in spelling performance can be explained through the stimulation of the lexical pathway: proficient readers and writers are therefore distinguished by high levels of use and effectiveness of the lexical or semantic route, therefore closely linked to improved efficiency of the “graphemic buffer” or graphemic store, which has the task of temporarily retaining the orthographic representations before these are converted into written language [28,29]. The automation of spelling rules and the consequent decrease in errors committed are closely linked to the strengthening of lexical competence.

This enhancement by increasing the knowledge of words stored in the phonologic, graphemic, and semantic storehouse, improves the child’s reading and writing skills and probably speeds up the transition from a predominantly indirect mode, typical of the inexperienced reader and writer, to a predominantly direct mode, typical of the effective reader and writer. The lexical route uses a direct link between orthographic memory and phonological memory of the word, in fact, the reader retrieves an articulatory pattern, stored stably in the lexicon, and does not have to be recreated each time by converting individual letters into individual phonemes.

Word learning through the semantic-lexical pathway is based on familiarization with words, listening to words in context, and repetition of those words in meaningful situations. Therefore, the enhancement program may have offered more exposure to words and provided a context-based learning opportunity. This may have contributed to the improvement of reading and spelling skills in the experimental group. Craik and Tulving’s [30] research suggested that depth of processing, which involves significant and elaborative mental activities (such as focusing on the semantic meaning of words), plays a critical role in the encoding and subsequent retrieval of information from memory. The depth of processing appeared to have a substantial impact on the ability to retrieve the words later [30]. This is crucial for comprehension and accuracy in reading and spelling.

The enhancement of the semantic processing skills requires the child to possess the ability to understand the relationships between words and not the only meaning of words, enabling him to increase his comprehension skills [31,32]. The child, in fact, gradually discovers the meanings of words thanks to the fact that they do not appear in isolation but are inserted into a context: the deep encoding of a text is associated with the enrichment of the lexicon that enables the understanding of the meaning of words, but above all, of the links between words. Knowing a word in-depth, in its multiple meanings and in its links with other words, makes it possible to read it accurately, because it allows the automation of processes and the central and deep elaboration of the word, with minimal intervention of sub-lexical processes, in which mainly superficial modes of word processing and construction are active [33,34,35]. This competence makes it possible to write it correctly from an orthographic point of view (improved efficiency of the graphemic buffer). Children in the experimental group have, at the end of the enhancement, a better efficiency of the “graphemic buffer”, through increased storage capacity and faster access to words, which has the task of temporarily retaining orthographic representations before they are converted into written language.

Moreover, the reinforcement enhances the phonological working memory, which seems to be extremely relevant, especially in the early stages of learning to read and write when the child learns the rules of grapheme-phoneme conversion, thus enhancing the indirect or sublexical pathway. It is precisely at this stage that children learn stable associations between written signs and sounds and for reading they operate a process of encoding visual material, which implies retaining the information in phonological form in the articulatory loop for the time necessary to carry out synthesis and analysis. The relationship between phonological working memory and reading success is that the latter is strongly conditioned by the ability to remember and recognize the constituent parts of language and their correspondence with alphabetic written language signs [36]. This type of access is the most rapid and correct process of decoding the written word as quickly and correctly as possible and consequently is the most advanced level of development of the processes of reading.

The increase in known words, stored in the phonological, semantic, and orthographic components, makes more information about the correct form in which these words are to be coded available at the time the reading and writing task is tackled. This finding is easy to interpret given the nature of the enhancement material, which required the child to engage in the reading or writing of particularly complex words of the Italian language, such as the word homophones (lago/l’ago; lake/the needle), or of the presence of certain exercises concerning the breakdown and reordering of words according to roots, suffixes, etc. or the presence of exercises about the use of contextual linguistic criteria, such as categorizations, functionalizations, synonymy, and antonymy. The child, when faced with such ambiguous terms, is better able to resolve the disambiguation of the word when it is inserted in a salient phrasal context rather than when it is considered singularly.

Moreover, the proposed exercises also allow a reflection on words and their meanings that enables one to contextualize the lexicon in deep cognitive-linguistic processing, and also, through metacognitive reflection, allows children to reflect on their own learning methods and on the reason that made it easy or difficult to tackle the exercises.

Additionally, although according to the literature [20], around the age of seven, children’s lexical competence starts to play a significant role in their reading and writing abilities, the efficiency of the lexical pathway is not fully developed at this age. This longitudinal study also demonstrated that enhancing lexical competence has an effect on second-grade pupils.

According to the staged model of the development of reading skills proposed by Uta Frith [20], around the age of four or five, the grapheme-phoneme conversion mechanism predominates, in which words are recognized or reproduced on the basis of physical and graphic characteristics. While, only after the age of eight, is the visual orthographic and phonological lexicon consolidated and the child becomes an increasingly expert and capable reader (and writer) as new words enrich his lexical-semantic storehouse, at the level of orthographic coding-decoding and at the representational level [37]. Around the age of eight, Italian-speaking children begin to use the semantic-lexical route more efficiently with regard to reading ability, but this developmental transition is much more complicated for writing ability [38,39,40].

Finally, in children undergoing enhancement, there is an increase in the number of known words, which leads to a corresponding increase in the number of words of which the correct orthographic form is stored, through a joint stimulation of the phonological, orthographic, and lexical components, which are evolutionarily connected. Consequently, the joint stimulation of the phonological and the lexical-semantic pathway enabled the children to familiarize themselves with certain linguistic obstacles, demonstrating the effectiveness of the enhancement of comprehension, reading, and spelling accuracy.

### Limitations and Implications of the Research

This study has some limitations. Firstly, this enhancement program is specific for the Italian language, which is a language with transparent orthography. This implies that the results are not generalizable to other languages, such as languages with opaque orthography (e.g., English). It would be interesting to investigate whether the same study conducted in English could produce the same or different results. In addition, recent studies show that lexical skills are important for formalized literacy development even in bilingual minority-language children [41]. Further research could also test the effectiveness of the enhancement program on these populations where the societal language (L-2) is Italian. Secondly, this specific enhancement program was tested on children with typical development. Future research should investigate the effectiveness of this enhancement program on children with atypical development, such as children with specific learning disorders, language disorders, or hearing impairment. Indeed, previous studies [2] have shown that this enhancement program is more effective on children with some impairment at the baseline. In fact, the program, acting on enhancing the lexical pathway, makes word recognition with direct semantic access easier and more expeditious. Finally, another limitation was not measuring the home literacy conditions of the participants, in fact, the questionnaire for collecting background information did not contain a specific questionnaire for measuring this variable. As recent studies on both preschool [42,43] and school [44,45] populations suggest, a very important factor in the development of some literacy skills is the home literacy environment. It would be desirable to take this variable into account as a control.

Regarding the implications, this enhancement program can be considered a good instrument to use on all children present in the class. This allows action on all children to maintain integration in the class. In fact, a strength of the program is definitely that it is a tool that can be integrated with typical teaching and thus can be implemented collectively. In addition, the exercises are constructed so as not to be boring and do not require the child to apply repetitive and often unnecessary spelling rules, in fact currently, the didactic tools used to promote the development of lexical competence are often composed of memorization and repetition activities, which do not always allow for generalization in ecological tasks or for the maintenance of learning over time.

## Figures and Tables

**Figure 1 children-10-01416-f001:**
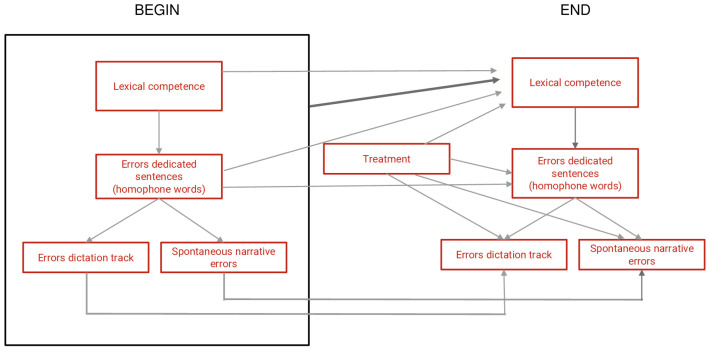
Multivariate analysis using graphical models. The rectangles show the variables studied to investigate a causal relationship. It should be noted that the treatment acts on all three types of errors and that there is an indirect effect of the treatment on errors in sentence dictation, which is through increased lexical competence (Reprinted/adapted with permission from Ref. [2]. Copyright year, copyright owner’s name) [2].

**Figure 2 children-10-01416-f002:**
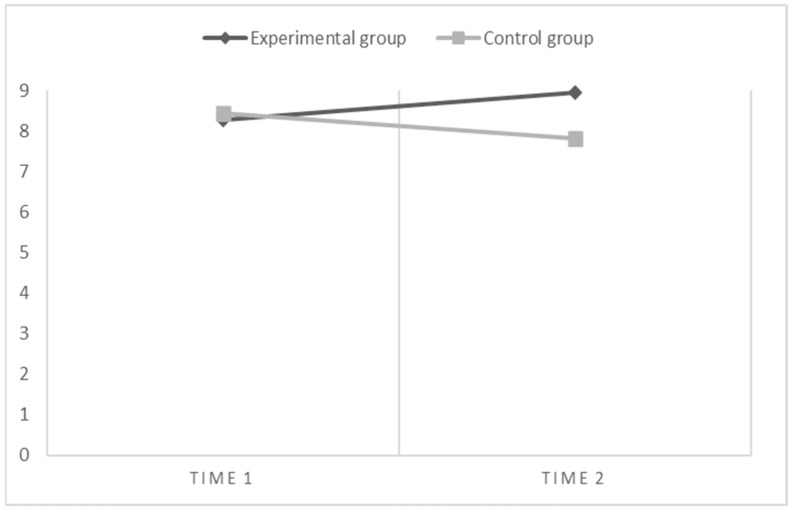
Graphical representation of effects when the dependent variable is Text Comprehension.

**Figure 3 children-10-01416-f003:**
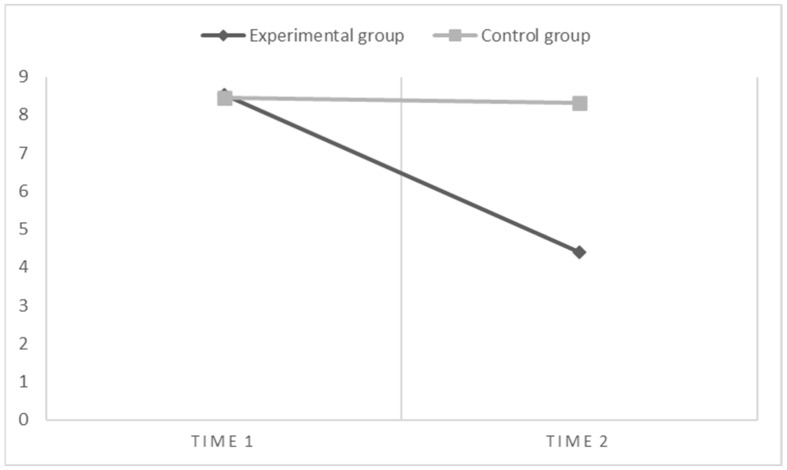
Graphical representation of effects when the dependent variable is Reading Inaccuracy (i.e., number of errors).

**Figure 4 children-10-01416-f004:**
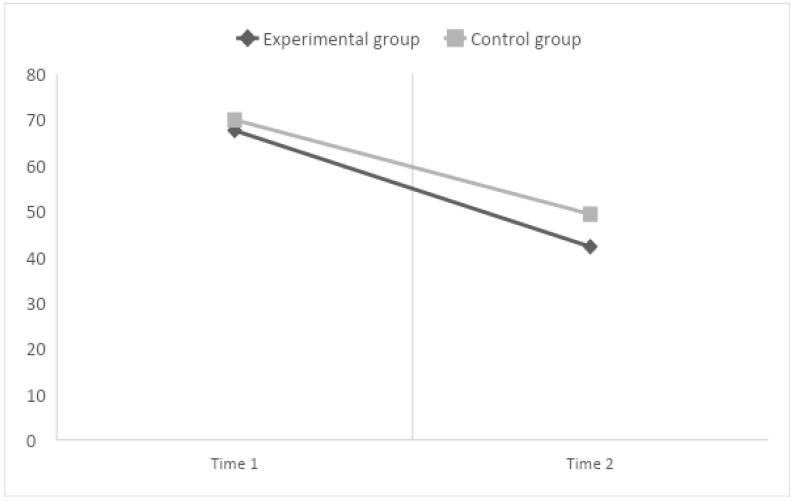
Graphical representation of effects when the dependent variable is Reading Time.

**Figure 5 children-10-01416-f005:**
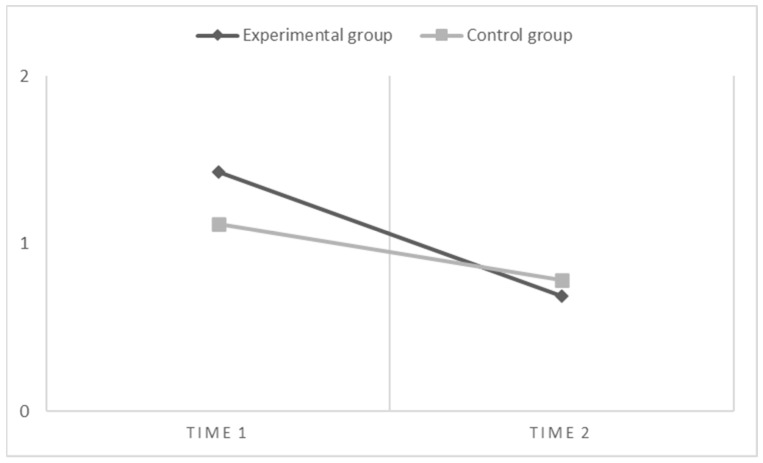
Graphical representation of effects when the dependent variable is Spelling Inaccuracy (i.e., number of errors).

**Table 1 children-10-01416-t001:** Main descriptive statistics (mean, standard deviation, minimum and maximum) and Levene’s statistics for homogeneity of variances, at the two times for experimental and control groups.

Variables	M (SD)		Minimum-Maximum		Levene’s Statistics
	Experimental group	Control group	Experimental group	Control group	
Text Comprehension					
Time 1	8.29 (1.83)	8.44 (2.05)	4–10	1–10	0.01
Time 2	8.97 (1.15)	7.82 (2.15)	4–10	2–10	15.39 ***
Reading Inaccuracy					
Time 1	8.53 (4.06)	8.46 (3.39)	2.5–19	2–15	0.35
Time 2	4.39 (3.15)	8.32 (4.80)	0–12	1.5–18.5	5.47 *
Reading Time					
Time 1	67.62 (20.88)	69.94 (34.85)	37.9–125	31.6–172.7	5.35 *
Time 2	42.22 (14.02)	49.34 (21.60)	27.2–82.2	21.1–114.3	4.11 *
Spelling Inaccuracy					
Time 1	1.43 (0.93)	1.12 (0.82)	0.2–4.4	0.1–3.6	0.19
Time 2	0.69 (0.60)	0.78 (0.61)	0.1–2.4	0–2.2	0.08

Note. If Levene’s test is significant, it means that the variances are not homogeneous, in the case it is not significant the variances are homogeneous. *** *p* < 0.001; * *p* < 0.05

**Table 2 children-10-01416-t002:** ANOVA statistics for comparison between experimental group and control group at Time 1.

Variables	M (SD)	Df	F Statistic	*p* Value
Text Comprehension				
Experimental group	8.29 (1.83)	1, 68	0.10	0.76 ^1^
Control group	8.44 (2.05)			
Reading Inaccuracy				
Experimental group	8.53 (4.06)	1, 68	0.01	0.94 ^1^
Control group	8.46 (3.39)			
Reading Time				
Experimental group	67.62 (20.88)	1, 68	0.11	0.75 ^1^
Control group	69.94 (34.85)			
Spelling inaccuracy				
Experimental group	1.43 (0.93)	1, 67	2.21	0.14 ^1^
Control group	1.12 (0.82)			

Note. Df indicates the degrees of freedom; ^1^ the *p* value is not significant.

## Data Availability

The data are not publicly available due to privacy.

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
