# Peer review of "Influence of Lexical Development on Reading and Spelling Skills: Effects of Enhancement on Second-Grade Children in Primary School"

_children, 2023, doi:10.3390/children10081416_

Round 1

Reviewer 1 Report

 The paper discusses the influence of the vocabulary in Italian on reading and spelling skills in second graders. The authors developed an intervention program to improve lexical acquisition which in turn is beneficial for reading and spelling. They tested their interventional program on second graders in Italian and compare their performance to an age-matched control group.

Overall I believe, that this study is very interesting and contributes well to the existing literature on other languages than Italian. However, the introduction needs major improvements and I have a few questions on the methodological part. The results are presented very clearly and the discussion fits well with the study. I will comment on the paper chronologically below: 

Title: I think that the title is a bit misleading. I believe the authors mean rather “enhancement on second-grade children in primary school”? and not an enhancement of the school itself.

Introduction: The first paragraph “This study … such as Italian.” is not a good start for an introduction and rather belongs in the abstract or at the end of the introduction as the aim/motivation of the study.

The authors start by citing themselves as previous research studies on the topic. I believe, that there are multiple further studies on other languages that can be cited here. Please elaborate and define the “relationship between lexical competence and encoding and decoding skills” further. Why is this important? 

Further the papers cited in 6 and 7 are very old. Since, the authors describe educational settings here, I hope there are more recent studies to cite on teaching vocabulary in class. Please cite these more recent studies at this point.

In the first lines on page 3 the authors mention quite briefly other studies that have shown the connection between lexical competence and spelling in English and adults. Why is this interesting to the reader here? Is there a link between the vocabulary and spelling or not? If the literature does not agree, then elaborate on the different perspectives please. If you mention a study then provide the research question, method and results please. Unless the readers cannot follow your argumentation. 

The authors write “This longitudinal study aims to see…” In my view this is not a longitudinal story, since they did not test the same children over several years. It is rather an interventional study. 

The sentence “who according to the literature regarding this age group tend to use the sub-lexical route more effectively.” – this reads as if the children consciously decide on one route or the other. I would not say that a cognitive process can be conscious. Please consider another wording that sounds less conscious.

In the next paragraph on page 3 the authors write “focusing more on quantitative (not qualitative) enrichment (number of words known).” The parentheses are confusing. The number of words known relates to quantitative enrichment. What does qualitative enrichment mean? Please provide an example as well. 

Participants: Please provide the number of male and female children not a percentage. The same goes for the split into two groups. It is unusual to use percentage. 

How were the groups chosen? Per school? Per class? Randomly? 

The authors mention that they collected background information of the participants. What were these characteristics and how were these used in the study (e.g., excluding children)?

Methods: Please provide for each subtest a time duration. 

Spelling task: How were the spelling errors rated? Did 2 errors in the same word count as 2 or 1? What does “balanced for total number of written words” mean?

Research design: see my comment above – it is not a longitudinal study, but an intervention.

Results: page 6 – please provide the p-value even if it is not significant. Table 2 needs some revision, please write the F-statistic as you did in the text on page 7 above Figure 2. 

Page 7 above Figure 3: the phrasing “the experimental group significantly decreased the number of errors made in the reading task” again sounds like a conscious decision. This needs a different wording. 

Discussion: page 9 last paragraph “The increase in the semantic system” – What does this mean? Please rephrase. 

Page 9 last sentence: What do the authors mean with “improved efficiency of the graphemic buffer”? Are more words stored or is the process of accessing these words faster? 

I like the fact that the authors included a paragraph about limitations of the study. They mention the home literacy environment as an interesting factor. At this point I was asking myself what background information was collected from the parents? Couldn’t the authors infer something from the parent survey for the literacy environment? Please elaborate.

Author Response

General comment

The paper discusses the influence of the vocabulary in Italian on reading and spelling skills in second graders. The authors developed an intervention program to improve lexical acquisition which in turn is beneficial for reading and spelling. They tested their interventional program on second graders in Italian and compared their performance to an age-matched control group.

Overall I believe that this study is very interesting and contributes well to the existing literature on other languages than Italian. However, the introduction needs major improvements and I have a few questions on the methodological part. The results are presented very clearly and the discussion fits well with the study. I will comment on the paper chronologically below: 

Authors’ response

Thank you for the review done to improve our work. Below we include the comments regarding the revisions made. In the text, changes have been highlighted to be immediately visible to the reviewer.

Comment #1

Title
I think that the title is a bit misleading. I believe the authors mean rather “enhancement on second-grade children in primary school”? and not an enhancement of the school itself.

Authors’ response

As suggested, we have changed the title.

Location: Title

Comment #2

Introduction

The first paragraph “This study … such as Italian.” is not a good start for an introduction and rather belongs in the abstract or at the end of the introduction as the aim/motivation of the study.

Authors’ response

Following the reviewer’s advice, we have reformulated the incipit of the introduction and moved the paragraph indicated at the end of the section to anticipate the aim and hypothesis of our study.

Location: Introduction, pages 3-4

Comment #3

The authors start by citing themselves as previous research studies on the topic. I believe, that there are multiple further studies on other languages that can be cited here. Please elaborate and define the “relationship between lexical competence and encoding and decoding skills” further. Why is this important? 

Authors’ response

Following the reviewer’s suggestion, we added recent studies investigating the relationship between lexical competence and encoding and decoding skills and the importance of this relationship. Also, in the light of the other comments, we more explained the characteristics of recent studies mentioned.

Location: Introduction, pages 2-3

Comment #4

Further the papers cited in 6 and 7 are very old. Since, the authors describe educational settings here, I hope there are more recent studies to cite on teaching vocabulary in class. Please cite these more recent studies at this point.

Authors’ response

We agree with the reviewer: we have cited more recent studies: 

[15] Loftus-Rattan et al., 2016;

[16] Seven et al., 2019.

Location : Introduction. Page 3

Comment #5

In the first lines on page 3 the authors mention quite briefly other studies that have shown the connection between lexical competence and spelling in English and adults. Why is this interesting to the reader here? Is there a link between the vocabulary and spelling or not? If the literature does not agree, then elaborate on the different perspectives please. If you mention a study then provide the research question, method and results please. Unless the readers cannot follow your argumentation. 

Authors’ response

To address all the concerns raised by the reviewer we deeply described studies cited above, highlighting how, in this type of sample, is confirmed the positive relationship between lexical competence and spelling. These mentions used to highlight the novelty of our study conducted on monolingual typically developing children.

Location: Introduction, page 3

Comment #6

The authors write “This longitudinal study aims to see…” In my view this is not a longitudinal story, since they did not test the same children over several years. It is rather an interventional study. 

Authors’ response

Thanks for this observation. We agree with the reviewer that this is an intervention study. However, as some statistical guidelines suggest, we can consider our study a "pre-post longitudinal study", therefore we have added this wording to specify the extent of the longitudinal study that we propose.

Below is an extract from the guidelines developed by researchers at Duke Global Health Institute: 

“The pre-test/post-test (‘pre-post’) design is one of simplest forms of longitudinal studies (Figure 1). The study often consists of a single baseline measurement, Yt=0, and is compared to a single follow-up measurement, Yt=1, usually occurring after an intervention.”

In support, we indicate other references:

Fitzmaurice, G.M., N.M. Laird, and J.H. Ware, Applied longitudinal analysis. 2nd ed. Wiley series in probability and statistics. 2011, Hoboken, N.J.: Wiley. xxv, 701 p. 

Liang, K.-Y. and S.L. Zeger, Longitudinal data analysis of continuous and discrete responses for pre-post designs. Sankhyā: The Indian Journal of Statistics, Series B, 2000: p. 134-148.

Location: Across main text

Comment #7

The sentence “who according to the literature regarding this age group tend to use the sub-lexical route more effectively.” – this reads as if the children consciously decide on one route or the other. I would not say that a cognitive process can be conscious. Please consider another wording that sounds less conscious.

Authors’ response

We agree with the reviewer and as suggested, we reformulated and provided a more detailed explanation to focus on the observed tendency without suggesting active choices on the part of children.

Location: Introduction, page 3

Comment #8

In the next paragraph on page 3 the authors write “focusing more on quantitative (not qualitative) enrichment (number of words known).” The parentheses are confusing. The number of words known relates to quantitative enrichment. What does qualitative enrichment mean? Please provide an example as well.

Authors’ response

Following the reviewer's advice, the sentence has been reformulated. To clarify, we removed the parentheses, and we provided a more detailed description of qualitative enrichment. 

This argument was introduced in this section but was further articulated in the enhancement program description and discussion.

Location: Rationale and aims, page: 4

Comment #9

Participant

Please provide the number of male and female children not a percentage. The same goes for the split into two groups. It is unusual to use percentage. 

How were the groups chosen? Per school? Per class? Randomly? 

Authors’ response

As suggested, we provided the description of the sample in terms of frequencies (n) and not of percentages.

Furthermore, we specified the method of recruitment of the subjects in the two groups of interest.

“The participants were selected by balancing the classes: for each school, an experimental class and a control class.”

Location: Participants, page 4

Comment #10

The authors mention that they collected background information of the participants. What were these characteristics and how were these used in the study (e.g., excluding children)?

Authors’ response

Following this observation, we better specified what kind of information was collected and why.

“Specifically, information relating to the subjects' environment was collected, such as for example the place of birth and the level of education of both parents, to verify the homogeneity of the groups in terms of socio-environmental characteristics. The collection of information showed that the subjects have socio-environmental characteristics of the middle class, and coming from the same catchment area, belong to a homogeneous population.”

Location: Participants, page 5

Comment #11

Methods

Please provide for each subtest a time duration.

Authors’ response

As suggested, we provided a time duration for each subtest.

Location: Section: Measures and Enhancement program, pages: 5-6

Comment #12

Spelling task

How were the spelling errors rated? Did 2 errors in the same word count as 2 or 1? What does “balanced for total number of written words” mean?

Authors’ response

Following this comment, we added an explanation of error coding, pointing out that errors were actually counted for the errors made in each word. 

In addition, we provided an explanation regarding the final score, which was calculated as the ratio of the number of errors made to the number of words written.

Location: Section: Measures and Enhancement program, page: 6

Comment #13

Research design

see my comment above – it is not a longitudinal study, but an intervention.

Authors’ response

Please, see reply to previous comment (Comment #6)

Comment #14

Results

page 6 – please provide the p-value even if it is not significant. Table 2 needs some revision, please write the F-statistic as you did in the text on page 7 above Figure 2. 

Authors’ response

According to the reviewer’s suggestion, we provide the p-value in Table 2, even if it is not significant. 

Furthermore, we have adjusted the errors on degrees of freedom (DF) in the table as they were reported with the period (.) and not with the comma (,).

Location: Results, page: 8

Comment #15

Page 7 above Figure 3

the phrasing “the experimental group significantly decreased the number of errors made in the reading task” again sounds like a conscious decision. This needs a different wording. 

Authors’ response

Following this suggestion, we modified the sentence in a different wording, as follows:

“These results suggest that after the enhancement program, the performances of the experimental group improved. Indeed significantly decreased the number of errors made in the reading task, compared to their peer control group.”

Location: Results, page: 9

Comment #16

Discussion

page 9 last paragraph “The increase in the semantic system” – What does this mean? Please rephrase. 

Authors’ response

As suggested, the sentence was reformulated. As follows:

“The enhancement of the semantic processing skills requires the child to possess the ability to understand the relationships between words and not the only meaning of words, enabling him to increase his comprehension skills”

Location: Discussion, page: 11

Comment #17

Page 9 last sentence: What do the authors mean with “improved efficiency of the graphemic buffer”? Are more words stored or is the process of accessing these words faster?

Authors’ response

According to the reviewer's advice, the sentence has been made more explicit. As follows:

“Children in the experimental group have, at the end of the enhancement, a better efficiency of the “graphemic buffer”, through increased storage capacity and faster access to words, which has the task of temporarily retaining orthographic representations before they are converted into written language.”

Location: Discussion, page: 12

Comment #18

I like the fact that the authors included a paragraph about limitations of the study. They mention the home literacy environment as an interesting factor. At this point I was asking myself what background information was collected from the parents? Couldn’t the authors infer something from the parent survey for the literacy environment? Please elaborate.

Authors’ response

Thank you for this observation. One of the limitations of our study was that information on the home literacy environment was not collected and for this reason it was further specified in the limitations section, with the prospect of being able to integrate this information in future studies.

We also specified, thanks to the previous comment, in the participants section which background information was collected through the questionnaire attached to the informed consent completed by the parents.

Location: Participants and Discussion (Limitations and implications of the research), pages: 5 and 13

Reviewer 2 Report

Dear authors, please find the review of your article attached.

Yours sincerely

Author Response

Thank you for the review done to improve our work. Below we include the comments regarding the revisions made. In the text, changes have been highlighted to be immediately visible to the reviewer.

Comment #1

The summary is well structured. It provides the information needed to understand the study.

The reader learns that the study is longitudinal, which is interesting. Perspectives are mentioned.

Authors’ response

Thank you for this feedback, we are glad you appreciated the structure of the abstract.

Comment #2

In the introduction, the authors state the aim of the study. We understand that this research question has already been explored, but not with 7-year-olds whose mother tongue is transparent. If this is the case, all the studies cited below should specify: the age of the children concerned AND the study language.

Authors’ response

Following the reviewer's comment, for each study cited we provided both the age of the participants and their language.

Location: Introduction, pages 2-3          

Comment #3

Furthermore, what theoretical criteria do the authors use to justify the age of interest, i.e. 7 years? What is significant about this age? Why would different results be expected in comparison with 9-year-olds?

Authors’ response

As suggested, the theoretical criteria used for the choice of this sample has been added and better specified. As follows:

“It was deemed appropriate to choose this age group for the development of lexical competence precisely to investigate what happens in the very early stages of learning written language. When children attend the first two years of primary school, they still exhibit those spelling and reading errors of the less developed type because they have not yet automated the competence of reading and writing. The theoretical criteria for justified this age of interest lies in Coltheart’s dual route model [18, 19] and in Firth’s staged model of the development of reading skills [20].  This staged model [20] suggests that at around four or five years old, children with a transparent language predominantly rely on the grapheme-phoneme conversion mechanism for word recognition, in line with Coltheart's sub-lexical route. From the age of eight, the visual orthographic and phonological lexicon are consolidated, leading to more proficient reading and writing as the child's lexical-semantic storehouse expands with new words.”

Location: Introduction, page 3

Comment #4

Finally, why specify the degree of transparency as a methodological criterion? Would the same study conducted in English have produced different results?

Authors’ response

Thank you for this observation. To clarify our motivation on the language’s transparency, we provided a specific description of criterion in Introduction section and, in addition, we added a future perspective in the Limitations and implications of research section.

For example: “This staged model [20] suggests that at around four or five years old, children with a transparent language predominantly rely on the grapheme-phoneme conversion mechanism for word recognition, in line with the Coltheart's sub-lexical route.”

“This implies that the results are not generalizable to other languages, such as languages with opaque orthography (e.g., English). It would be interesting to investigate whether the same study conducted in English could produce the same or different results.”

Location: Introduction and Limitations and implications of research, pages 4 and 13

Comment #5

“this competence and vocabulary enrichment is strictly dependent on chance opportunities provided by listening and speaking situations, teaching practice or reading activities [3].” ? This sentence seems to me to be badly formulated. Before entering school, or at least before formal language learning, this skill is indeed acquired implicitly through repeated contact with the language elements of the environment, but this is no longer STRICTLY the case afterwards... In France, the curricula insist on developing lexical skills at school in cycles 1, 2 and 3. This involves specific teaching sequences.

Authors’ response

As suggested, the concept was better specified. In the preschool period these skills are strictly dependent on listening and speaking opportunities in the family context, and how important teaching practices formally in use are instead in the formal literacy period.

Location: Introduction, page 3

Comment #6

“The complex connections between lexical competence and reading skills or reading comprehension are confirmed by recent literature; however, studies have focused mainly on language with opaque orthography [9, 10]”. And why should this be specific to transparent languages? The authors should continue the argumentation by unrolling their red thread and clearly spelling out the arguments in favor of this study. It's not up to the reader to make the inferences.

Authors’ response

As suggested, we modified this sentence, and we added more studies conducted in primary school children with transparent languages.

“The complex connections between lexical competence and reading skills or reading comprehension are confirmed by recent literature. Subjects with a strong vocabulary foundation find it easier to comprehend written texts, infer meanings from context, and express their ideas in writing and also have a tendency to show better reading comprehension skills [3, 4].”

Location: Introduction, page 2

Comment #7

« However, in recent years the precise nature of the link between lexical competence and spelling accuracy is an area of ongoing investigation: ». Your study does not assess the NATURE of the link. Please specify.

Authors’ response

Following your suggestions, we have extensively reworded this paragraph, making our intentions clearer and more specific.

“On the other hand, regarding the relationship between lexical competence and spelling abilities, different studies have shown their connection in samples who are learning English as a second language [8, 9] or in adult samples [10]. In these studies, the importance of breadth, depth and productivity of lexical competence in their interaction with writing skills has been confirmed, highlighting how this competence is a good predictor of the quality of writing performance. Actually, the novelty of studies already mentioned by Bigozzi and colleagues [1, 2] lies in their emphasis on how lexical competence is fundamental to the development of spelling ability in monolingual and typically developing children, investigated these aspects in a sample of primary school children with an average age of about nine years, in a language with transparent orthography, such as Italian.”

Location: Introduction, page 3

Comment #8

“The teaching methods traditionally used to enhance lexical competence and promote vocabulary enrichment are structured with memorization activities of specific words, focusing more on quantitative (not qualitative) enrichment (number of words known)”.--> In Italy ?

Authors’ response

Following this observation, we have made it more explicit that we are referring to the Italian school system.

Location: Rationale and aims, page 4

Comment #9

In the participants section, you say “It was deemed appropriate to choose this age group for the development of lexical competence precisely to investigate what happens in the very early stages of learning written language. » Developmental aspects should be developed in the theoretical section. The reader should understand why this age group is relevant or suitable for the study without waiting until the Materials section. Recent theoretical references should also be provided.

Authors’ response

Following this suggestion, we decided to move this sentence in the Introduction to more explain why this age group is relevant fot the study.

Location: Introduction, page 3

Comment #10

Material

«All tests were administered in schools, during school hours.” Who administered the tests? By the experimenter? By the teachers? If so, this could be problematic. What precautions were taken?

Following the reviewer's remark, we specified that the tests, as well as the enhancement programme, were administered by an investigator who had been trained beforehand. This took place during school hours, after scheduling arrangements had been made with the teachers involved and the head teacher.

Location: Measures and Enhancement program, page 5

Comment #11

« Two independent raters coded the errors, the agreement between the raters was 98%, the disagreements were discussed and resolved. » Specify the statistical test used.

Authors’ response

Following statistical recommendations (Pharm, N. G., Bell, J. S., & Chen, T. F. (2013). Interrater agreement and interrater reliability: Key concepts, approach, and applications. Research in Social and Administrative Pharmacy, 9(3), 330-338), we specified in the text how the percentage index of agreement among inter-raters was calculated for the evaluation of the test.

Location: Measures and Enhancement program, pages: 5-6

Comment #12

Results

The control group went from 8.44 to 7.82 between the two times. How do you explain this drop? Is it significant? Please specify in the table.

In the table, we understand that you calculate the difference between the experimental group and the control group at each time. Is this correct? But it would be wiser to check the difference between the score at time 1 and at time for EACH group independently.

Authors’ response

Thanks for this observation. The result of understanding the text of the control group actually decreases at T2, but the data is irrelevant, not significant. This result is visible in the results reported below, conducted with the repeated measures ANOVA.

As it is possible to see from the captions, Table 1 refers only to descriptive statistics, the results of the Levene test do not measure the differences between T1 and T2 in the subjects but the homogeneity of the variance (therefore used as descriptive statistical data, as described in the text preceding the table).

Table 2, on the other hand, reports the differences at T1 between the two groups on each variable to verify that the participants (both in the control and experimental groups) both started from the same performance state and that therefore we could attribute the differences at T2 to the program of enhancement and not to initial differences.

We hope this explanation is clear enough.

Round 2

Reviewer 1 Report

Thank you for revising the manuscript and replying to my comments. The paper has improved immensly. Good job.

Reviewer 2 Report

Thank you